# The Relationship between Stress, Inflammation, and Depression

**DOI:** 10.3390/biomedicines10081929

**Published:** 2022-08-09

**Authors:** Il-Bin Kim, Jae-Hon Lee, Seon-Cheol Park

**Affiliations:** 1Department of Psychiatry, Hanyang University Guri Hospital, Guri 11923, Korea; 2Graduate School of Medical Science and Engineering, Korea Advanced Institute of Science and Technology (KAIST), Daejeon 34141, Korea; 3Department of Psychiatry, Schulich of Medicine and Dentistry, Western University, London, ON N6A 3K7, Canada; 4Department of Psychiatry, Hanyang University College of Medicine, Seoul 04763, Korea

**Keywords:** stress, chronic inflammation, depression, inflammatory cytokines, stress hormones, antidepressant, immune system, immunomodulatory effect

## Abstract

A narrative review about the relationship between stress, inflammation, and depression is made as follows: Chronic stress leads to various stress-related diseases such as depression. Although most human diseases are related to stress exposure, the common pathways between stress and pathophysiological processes of different disorders are still debatable. Chronic inflammation is a crucial component of chronic diseases, including depression. Both experimental and clinical studies have demonstrated that an increase in the levels of pro-inflammatory cytokines and stress hormones, such as glucocorticoids, substantially contributes to the behavioral alterations associated with depression. Evidence suggests that inflammation plays a key role in the pathology of stress-related diseases; however, this link has not yet been completely explored. In this study, we aimed to determine the role of inflammation in stress-induced diseases and whether a common pathway for depression exists. Recent studies support pharmacological and non-pharmacological treatment approaches significantly associated with ameliorating depression-related inflammation. In addition, major depression can be associated with an activated immune system, whereas antidepressants can exert immunomodulatory effects. Moreover, non-pharmacological treatments for major depression (i.e., exercise) may be mediated by anti-inflammatory actions. This narrative review highlights the mechanisms underlying inflammation and provides new insights into the prevention and treatment of stress-related diseases, particularly depression.

## 1. Introduction

The pathogenesis of major depression has been mainly explained by the neurogenesis theory for the purpose of overcoming the limitations of the monoamine theory. In terms of the neurogenesis theory, an interaction of serotonergic drugs with postsynaptic 5-HT receptors can be mediated by guanine-nucleotide-binding proteins. In addition, stimulated adenylate cyclase, accompanied by a G-protein, sequentially activates cyclic adenosine monophosphate (cAMP) and increases the level of protein kinase. In contrast, an interaction of noradrenergic drugs with postsynaptic norepinephrine (NE) receptors can inhibit adenylate cyclase. Herein, the second messenger system of 5-HT has a stimulatory effect, whereas that of NE has an inhibitory effect. In addition, increased brain-derived neurotropic factor (BDNF) can alleviate the imbalanced levels of 5-HT and NE in the condition of major depression. However, the other pathogenic process of major depression (i.e., inflammation) or the potential targets of antidepressants have been newly proposed by solving the unmet need for the pharmacological and non-pharmacological treatments of major depression [1,2,3]. Chronic stress can lead to a state of homeostasis dysregulation triggered by environmental and psychological stressors [4,5]. Stressful events activate the sympathetic nervous system and the hypothalamic–pituitary–adrenal (HPA) axis, which engenders multiple neurotransmitters, neurochemicals, and hormonal alterations. The sympathetic nervous system and the HPA axis release chemical mediators that protect the body from stress. Catecholamines increase heart rate and blood pressure, which trigger the “fight or flight” response. In the last two decades, accumulating evidence has indicated that severe stress increases the risk of physical and psychiatric disorders. These are known stress-related diseases. Stress is a common risk factor for over 75% of physical and mental diseases, increasing the morbidity and mortality of these diseases [6]. Psychiatric disorders such as depression are the most common stress-related diseases [6,7,8].

Conventional mechanisms linking stress and disease have focused on the HPA axis and the sympathetic nervous system. However, alterations in the HPA axis and the sympathetic nervous system mainly affect the immune system; thus, the mechanisms linking stress to stress-related diseases remain unclear. Inflammation has emerged as a key mechanism underlying stress [9,10,11,12,13]. Additionally, immune imbalance, especially reflected in reduced natural killer cell activity, has been associated with depressed mood. Furthermore, many studies have indicated that pharmacotherapy has resulted in augmented natural killer cell activity, such as improving depressed mood [14]. A review of the literature suggests that inflammation contributes to the pathophysiology of stress-related disorders. We hypothesized that inflammation is a common factor in stress-related diseases and attempted to elucidate the association between them.

## 2. Stress and Inflammation

There is evidence that stress can induce inflammatory changes in the brain and peripheral immune system [9,15,16,17,18,19]. Communication between the neuroendocrine and immune systems has been previously reported in different studies [20,21,22,23,24,25]. Stress activates the HPA axis. The hypothalamic secretion of corticotropin-releasing hormones suppresses immune responses by mediating the release of glucocorticoids from the adrenal glands. Studies have revealed that glucocorticoids inhibit lymphocyte cytotoxicity and proliferation [26,27]. Furthermore, they reduce the expression of several pro-inflammatory cytokines, such as interleukin (IL)-6 and tumor necrosis factor (TNF)-α, and they increase the expression of anti-inflammatory cytokines (such as IL-10 and TNF-β) [28,29,30,31,32,33]. However, glucocorticoids also have a pro-inflammatory effect on the immune system [34,35,36,37,38,39]. They enhance the function of the inflammasome NLRP3 through an increase in IL-1β secretion in response to adenosine triphosphate. Inflammasomes are multiple protein complexes that sense external and internal hazardous signals, leading to the cleavage of pro-inflammatory cytokines into mature cytokines, including IL-1β and IL-18. One study revealed that the pro-inflammatory role of glucocorticoids comprises the activation of the innate immune system in response to hazard signals [40,41]. Pro-inflammatory factors, including IL-1, IL-6, and TNF-α, activate the pituitary–adrenal axis, leading to increased serum levels of adrenocorticotropic hormone (ACTH) and glucocorticoids, which, in turn, inhibit the production of pro-inflammatory factors [42,43,44,45,46,47,48]. The immune system and the HPA axis interact to form a negative feedback mechanism. Negative feedback loops can be impaired by a decrease in the expression of cytoplasmic glucocorticoid receptors (GRs) and GR-driven anti-inflammatory genes, resulting in reduced responsiveness to glucocorticoids when stressors overstimulate cytokines [49,50]. In addition to glucocorticoids, the sympathetic nervous system and related neurotransmitters, such as NE and neuropeptide Y (NPY), can affect the immune system and inflammatory function. NE increases the phosphorylation of mitogen-activated protein kinases (MAPKs) through an α-receptor-dependent pathway that activates the secretion of inflammatory factors. Additionally, NPY activates the production of transforming growth factor-β (TGF-β) and TNF-α in macrophage-like cells via the Y1 receptor [51,52,53,54,55].

Both pro- and anti-inflammatory systems are influenced by the intensity and type of stressors [56]. Acute stressors increase immune function, whereas chronic stressors suppress it. Excessive stressors overstimulate the immune system, producing an imbalance between pro- and anti-inflammatory effects. Studies have reported that pro-inflammatory upregulation is induced by stress factors, such as C-reactive protein (CRP), IL-1β, IL-6, TNFα, and nuclear factor kappa B (NF-κB) [57,58,59,60,61,62].

In addition to peripheral inflammation, neuroinflammation has also been reported in response to various stressful conditions [18,63,64,65,66,67,68,69,70]. Pro-inflammatory cytokines that activate microglia lead to the peripheral accumulation of monocytes and macrophages. They have also been found in the brain after stress exposure [71,72,73,74,75,76,77]. Microglia, which are macrophages in the brain, are regarded as the principal source of pro-inflammatory cytokines. Microglia express both glucocorticoid and mineralocorticoid receptors and directly respond to a peak in corticosterone [15,16,78,79]. Additionally, GRs are highly concentrated in the prefrontal cortex and the hippocampus. Thus, stress-related corticosterone indirectly affects microglia. A recent study showed that the innate immune system in the CNS responds to acute stressors by releasing the hazard signal high-mobility group box-1 (HMGB-1) in primate brains, which activates microglia by acting on the NLRP3 inflammasome [80]. Activated microglia exhibit a change in morphology, which is observed as a hypertrophied soma with an enlarged branch. This leads to an increase in cytokines that recruit monocytes from the periphery. An increase in macrophages and peripheral monocytes promotes the production of pro-inflammatory cytokines, such as IL-1β, TNFα, and IL-6, in the brain [81].

Common processes include the activation of the immune system, an increase in sympathetic nervous system pathways, and a decrease in the responsiveness of glucocorticoids, causing the activation of inflammatory responses to stress. Catecholamines, cytokines, glucocorticoids, and other mediators secreted in response to stressors may be the main players in stress-related pro-inflammatory changes.

## 3. Stress, Inflammation and Depression

Traditionally, inflammation has been considered an essential response to tissue injury or microbial invasion in order to protect homeostasis. Although other pathogeneses are being actively investigated, the molecular basis of the inflammatory pathway is considered critical in the pathogenesis of psychiatric disorders, including depression [82,83,84,85,86]. Increasing evidence suggests that excessive inflammation plays a crucial role in the onset and progression of stress-related diseases. Evidence supports the idea that the inflammatory response forms the background of multifactorial diseases, including psychiatric disorders [13,87,88,89,90].

Stressful events are elemental in inciting episodes of major depressive disorder (MDD). Patients with depression are predisposed to the activation of the HPA axis and hypercortisolemia, rising levels of stress hormones and corticotropin-releasing hormone, and an increase in ACTH [91,92,93,94]. MAPKs enhance the activity of serotonin membrane transporters, and serotonin is the most vital neurotransmitter related to depression [95,96,97,98].

Various concepts of inflammation, such as the macrophage hypothesis and the cytokine theory, have been suggested as the underlying pathology in MDD. The main principle of inflammatory depression is the activation of the immune response, particularly cytokine production. This affects the levels of neurochemicals that lead to MDD [99]. Stress can promote the development of depression-like behavior by facilitating the expression of inflammatory cytokines [100,101]. Recently, the novel kynurenine pathway (KP) has gained attention in the cytokine theory. Pro-inflammatory cytokines activate KP to influence tryptophan metabolism and secrete neurotoxins, which may either decrease serotonin production or promote serotonin reuptake [102,103].

Studies on patients and animal models have supported inflammation as a pathology of depression. Inflammatory cytokines, such as IL-1β, TNF-α, and IFN-α, or inducers, such as vaccinations, have been found to result in alterations in human and rodent behaviors. Increased levels of inflammatory mediators, including cytokines, acute phase proteins, chemokines, adhesion molecules, and prostaglandins, have also been found in the CNS, blood, and cerebrospinal fluid of patients with depression [57,100,104,105,106,107]. Chronic stress is frequently used to establish depression models. Exposure to chronic stress for four weeks significantly activates inflammatory cytokines, including IL-18, IL-1β, TNF-α, and inflammatory inducible nitric oxide synthase (NOS) [108]. With the upregulation of pro-inflammatory cytokines, depression-like behaviors are provoked. Blocking inflammatory cytokines or inducible NOS with agents such as minocycline helps eliminate the depression-like behavior caused by stress [108]. Several antidepressants have been shown to have anti-inflammatory effects. Antidepressants and nonsteroidal anti-inflammatory drugs, such as minocycline, mitigate microglial activation and reduce blood levels of IL-6 and central cytokine secretion, thereby diminishing behavioral alterations [109].

Inflammasomes are multiple protein complexes that drive the production and maturation of pro-inflammatory factors, such as IL-18 and IL-1β, to induce innate immune defenses [110,111]. The NLRP3 inflammasome has been implicated in the induction of depression-like behaviors in a lipopolysaccharide (LPS)-induced mouse model [112,113]. A recent study demonstrated that the protective role of caspase-1 inhibition in brain function and gut microbiota provoked depression-like and anxiety-like behaviors [114,115,116,117].

## 4. MDD Is Associated with an Activated Immune System

MDD is associated with diseases characterized by an activated immune system, such as autoimmune diseases (systemic lupus erythematosus (SLE), type 1 diabetes, and rheumatoid arthritis (RA)), allergies, and infections (sepsis). Patients with both asthma and atopy have an approximately 50% higher incidence of MDD [115,116]. Approximately, 36% of patients with asthma suffer from MDD, and their TNFα levels are significantly increased, while their IFNγ levels are significantly decreased.

Meta-analyses have revealed that the incidence of MDD in patients with diabetes is up to twice that in those without diabetes [118,119]. Inflammatory activation is linked to the pathogenesis of diabetes, with the immune response implicated in both type 1 and type 2 diabetes [120]. Inflammatory markers, including CRP, IL-1β, IL-1RA, and MCP-1, are significantly increased in patients with MDD and type 2 diabetes [121].

One-third of the patients with SLE were reported to suffer from MDD in another meta-analysis that used the standard depression scale subscale for depression and hospital anxiety [122]. Studies have also shown that severe fatigue is associated with a high risk of MDD and that there is no correlation with SLE severity [123,124]. A review found that over 90% of patients with SLE experience fatigue without an association with the severity of SLE [125]. Increased TNF-α and decreased IL-10 levels have been demonstrated in patients with SLE and MDD and have been associated with a more severe course of MDD [126,127].

MDD is also more prevalent in patients with RA. Studies have reported a 74% higher risk of RA in patients with MDD than in healthy controls. In addition, 17% of patients with RA have MDD [128,129]. More than 70% of patients with RA experience clinically significant fatigue [130]. A study reported a positive association between serum CRP levels and MDD severity among patients with RA [131]. CRP levels and the erythrocyte sedimentation rate, a marker of inflammation severity, are significantly associated with clinically significant fatigue [132]. A review investigated diverse anti-TNF and other biological agents in patients with RA and reported that they had significant impacts on fatigue felt by patients, further demonstrating that fatigue may partly underpin immune responses [133].

The association between immune activation and MDD has not only been reported in immune-related disorders but also in cases of infections. Sepsis leads to broad pro-inflammatory reactions that trigger a systemic immune response to an infective agent. Patients with sepsis have a substantially increased level of inflammatory markers, even after sepsis resolution [134,135,136]. Sepsis survivors have an increased rate of MDD compared to normal controls; however, this has been found to not be significantly higher than the rates of MDD preceding infection [137]. This high rate of MDD in patients with sepsis also reflects that psychological stress increases MDD and immune activation [138] and is related to a higher risk of sepsis [139]. Few studies have reported post-sepsis MDD in human patients, while animal studies have reported sepsis conditions leading to changes in emotion [140]. Animal studies have also reported that immune suppression, by suppressing the NF-κB pathway, reduces MDD-like behaviors in animal models [140,141]. There may be a possible role for the “priming” of the immune response by conditions such as sepsis, in turn triggering a higher risk of developing MDD [142]. Future studies are warranted to elucidate whether immune activation predisposes the immune system to be more reactive to stress and insults, resulting in an increased risk of MDD.

Recently, a bioinformatics analytical approach has gained prevalence using peripheral blood biomarkers by screening for potential mRNAs in patients with MDD [143]. Zhang et al.,used public data from an mRNA microarray to screen for hub gene alterations implicated in MDD. Differentially expressed genes were analyzed using the Kyoto Encyclopedia of Genes and Genomes and Gene Ontology pathway datasets. A combined gene set of co-upregulated differentially expressed genes was associated with immune response, neutrophil activation, degranulation, cell-mediated immunity, and the Toll-like receptor signaling pathway.

## 5. Antidepressants Exert Immunomodulatory Effects

The response to antidepressants has been associated with changes in immune marker levels. In animal models treated with LPS, low serum levels of TNFα and high levels of IL-10 were identified following the administration of an serotonin reuptake inhibitor (SSRI) and a serotonin–norepinephrine reuptake inhibitor (SNRI) [144]. In a social stress model, tricyclic antidepressant (TCA) administration reduced microglial IL-6 mRNA in vivo and ex vivo and decreased TNF-α and IL-1β mRNA levels [145]. Studies using macrophages in animal models have also found a similar immunosuppressive influence, in which the reduction in IL-6 and enhancement in IL-10 levels following antidepressant administration indicated that such impacts might be mediated by the inhibition of the NF-κB pathway [146]. Meanwhile, a study revealed that the administration of SSRIs and mirtazapine resulted in the opposite effect on cytokine production, with increased levels of inflammatory markers, including IL-6, IL-1β, and TNF-α [147]. Human studies investigating alterations in cytokine levels have found that the administration of antidepressant agents decreased the levels of IL-1β, IL-4, IL-6, and IL-10 [148,149]. Other studies have shown that antidepressants have different immunomodulatory effects. SNRIs, such as venlafaxine, have a higher anti-inflammatory effect than SSRIs [150]. This study also demonstrated that SSRI administration significantly enhanced serum IL-6 and TNFα levels. Other studies have also found that psychotherapy has an immunomodulatory effect similar to that of pharmaceutical treatment [151]. Recent studies focused on physical exercise and transcranial direct current stimulation have reported that circulating cytokine levels decreased following treatment. However, there are no consistent findings regarding the improvement in MDD symptoms [152,153]. The administration of electroconvulsive therapy (ECT) has a similar effect on the immune system. ECT is linked to an initial increase in IL-1 and IL-6, with the levels of TNFα and IL-6 decreasing after long-term treatment. However, these results need to be further replicated [154]. One study focused on the effect of ECT as an adjunctive to antidepressant agents and reported that, while it significantly reduced IL-6 levels, TNFα levels increased after treatment [155]. ECT has also been reported to reverse changes in NK cell activity, which is decreased in patients with MDD [156].

Studies have also demonstrated that immune markers can be used to predict therapeutic efficacy. Low levels of pro-inflammatory cytokines can predict better therapeutic responses to SSRIs, TCAs, and ketamine, and responders may have significantly reduced cytokine levels [157,158]. However, serum CRP levels predict different therapeutic responses to different antidepressant agents [159]. Patients with reduced CRP levels showed a better response to escitalopram, while those with increased CRP levels demonstrated a better response to TCA, such as nortriptyline. These findings suggest that SSRI effects may be partly derived from anti-inflammatory actions. High IL-6 levels in patients have also been correlated with worse therapeutic efficacy of different SSRIs and SNRIs [160]. However, higher TNF-α levels were correlated with therapeutic non-response in patients treated with escitalopram [161].

In sleep deprivation therapy, an antidepressant treatment option, high IL-6 levels predict worse therapeutic responses in patients with MDD [162]. Reduced TNF-α levels at the first ECT have also been shown to predict therapeutic outcomes [163]. However, the link between high levels of inflammatory cytokines and poor therapeutic efficacy has not been confirmed for all therapeutic options. It has been demonstrated that high pro-inflammatory cytokine levels, such as TNFα, predict a favorable response to physical exercise [153]. The differences in the predictive effects of inflammatory cytokine levels in light of the different therapeutic efficacies indicate that their underlying mechanisms may differ, with anti-inflammatory influences being more crucial for some therapeutic options, such as SSRIs, than others.

Fatigue may be effectively controlled with a few medications. However, few studies have investigated the association between these medications and the immune system. Amantadine can be effective in patients with multiple sclerosis [164]; however, little research has focused on its immunomodulatory effects. Amantadine administration in rats enhanced the effect of fluoxetine, an SSRI, when co-administered; however, it did not alter IFNγ or IL-10 levels [165]. Further studies are warranted to examine whether its therapeutic efficacy for fatigue in patients may act via its effects on the immune system.

## 6. Non-Pharmacological Treatment Effects on Inflammatory Depression

Omega-3 fatty acids have been identified as potential treatments for MDD-related inflammations. Patients with MDD have low levels of n-3 polyunsaturated fatty acids (PUFA) [166], indicating that n-3 PUFAs may be involved in MDD pathophysiology. Most randomized controlled trials investigating the antidepressant effects of n-3 PUFAs have shown small-to-medium effects [165,166,167,168,169], as only patients with signs of low-grade inflammation responded to this PUFA intervention. This may be consistent with a study showing that n-3 PUFA administration hinders LPS-induced depression-like behavior in mice via the suppression of neuroinflammation [170]. Another study reported that n-3 PUFA administration had a protective effect on IFN-α-induced MDD, further supporting the role of n-3 PUFA in MDD-related inflammation [171]. Exercise therapy may be an appropriate alternative to conventional antidepressants [172,173,174]. Interestingly, recent studies have shown that the antidepressant effects of physical exercise may be mediated by anti-inflammatory actions [175]. A clinical study demonstrated that patients with MDD who partially responded to SSRIs showed a significant decrease in TNF-α levels after 12 weeks of physical exercise [153].

## 7. Limitations and Future Directions

The literature regarding stress-related inflammation reviewed here may be useful for understanding the common physiology of stress-related diseases. However, there are unanswered questions that require further investigation. In addition to inflammation, crosstalk between inflammation and other related pathways, such as cell stress, has not been extensively investigated. Moreover, there have been very few studies on the correlation between specific pathways and stress-related diseases. It is unknown whether other pathways or mechanisms that do not modify the peripheral immune system can affect neuroinflammation in depression. From a clinical perspective, anti-inflammatory treatment must target specific cells and pathways in the brain that are crucial in the pathogenesis of depression. Further research is required to address these limitations, which will contribute to the development of novel therapies for stress-related diseases, particularly depression. To improve stress, psychological and physical stressors related to multifactorial mechanisms should be prevented in patients with depression. Moreover, targeting stress-related risk factors comprising stress-induced inflammation could be beneficial in preventing diseases among highly stressful individuals, such as those with depression.

## 8. Conclusions

Stress prompts inflammation not only peripherally but also centrally through the dysregulation of the immune system (Figure 1). This results in the development of stress-related diseases, particularly depression. Although there are various facilitating factors, inflammation appears to be the central pathology that leads to depression. In this review, we provide evidence that stress induces depression through neuroinflammation and peripheral inflammation. Stress weakens the central microglia, blood, and immune system by activating the sympathetic nervous system and the HPA axis. Thus, our findings support the notion that inflammation is the main pathway for stress-related diseases, such as depression, and that it contributes to disease progression by affecting the early course of the disease. The altered activity of the hypothalamic–pituitary–adrenal (HPA) axis, the weakened blood–brain barrier (BBB), and the facilitated passage of pro-inflammatory cytokines may be related to the serotonergic, noradrenergic, and dopaminergic dysfunction of major depression [176,177,178]. In addition, the tryptophan–kynurenine metabolic pathway is involved in the process of chronic neuroinflammation, which is associated with major depression, in the context of pro-inflammatory cytokines and the immune response [179,180]. Moreover, PUFAs have the same neuroprotective effects as non-pharmacological treatments for depression, based on their effects on microglial polarization [181].

## Figures and Tables

**Figure 1 biomedicines-10-01929-f001:**
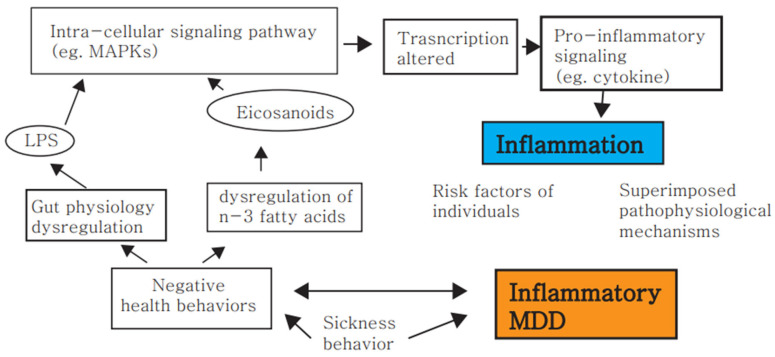
Assumptive model of reciprocal interactions between low-grade inflammatory changes and MDD. Negative health behaviors, including poor stress-coping strategies, may provoke inflammatory responses, which involve potential mechanisms mediating the effects of such negative behaviors on inflammation, dysregulation of n-3 fatty acids, and gut physiology dysregulation. This, in turn, may generate symptoms of MDD, including motivational deprivation and psychomotor impairment, further aggravating sickness behavior patterns, and may induce depression and more negative health behaviors. Abbreviations: omega-3 (n-3), lipopolysaccharide (LPS), mitogen-activated protein kinases (MAPKs), major depressive disorder (MDD).

## Data Availability

Not applicable.

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
