# Peer review of "The Relationship between Stress, Inflammation, and Depression"

_biomedicines, 2022, doi:10.3390/biomedicines10081929_

Round 1

Reviewer 1 Report

This Review focuses on complicated relationships between stress, inflammation and stress –related diseases like depression.

The Authors provide evidence that stress induces depression via neuroinflammation and peripheral inflammation and that stress weakens the central microglia, blood, and immune system by activating the sympathetic nervous system and hypothalamic-pituitary-adrenal axis. The key finding supports the notion that inflammation is the main pathway for stress-related diseases, like depression, which contributes to disease progression by affecting the early course of the disease.

This is a very important review enlightening a very hot topic. It definitely can be published.

There only several small points.

1 To attract more citations, please, extend and reorganize the Abstract.

Please, insert a short description of the effects of depression on the main pathways of immunity. Also. Please, underline principally new finding or analytic opening provided by the Review.

2  Please, Add several words to the Introduction about the effects of depression
of Natural Killer Cell Activity and underline NK importance in
anti-cancer immunity.

3 If it is possible, please, prepare a color figure or a Table illustrating and explaining depression-caused imbalance
 between pro- and anti inflammatory immunity

4 It also makes sense to review the latest Results obtained with Systems
biology and bioinformatic approaches.

5 If it is possible, please, add a small chapter describing modern non Pharmacological
 approaches to depression treatment, including Clinical Hypnosis, CBT,
 Exercises, Art and Music therapy and, anti-inflammatoty diet (containing
food with low histamine content)

Author Response

Reviewer 1

1 To attract more citations, please, extend and reorganize the Abstract.

Please, insert a short description of the effects of depression on the main pathways of immunity. Also. Please, underline principally new finding or analytic opening provided by the Review.

  • We followed your valuable advice. Please refer to the revised manuscript which has yellow shaded sentences.

2  Please, Add several words to the Introduction about the effects of depression of Natural Killer Cell Activity and underline NK importance in anti-cancer immunity.

  • We added the sentences describing the association between NKC and depression in introduction. Please refer to the revised manuscript that is highlighted with yellowed shaded sections.

3 If it is possible, please, prepare a color figure or a Table illustrating and explaining depression-caused imbalance between pro- and anti inflammatory immunity

  • We added a figure in the manuscript according to your valuable advice.

4 It also makes sense to review the latest Results obtained with Systems biology and bioinformatic approaches.

  • We added sentence describing a study on 2020 that used bioinformatic approach to screen inflammation, immunity-associated biomarkers of depression.

5 If it is possible, please, add a small chapter describing modern non Pharmacological approaches to depression treatment, including Clinical Hypnosis, CBT, Exercises, Art and Music therapy and, anti-inflammatoty diet (containing food with low histamine content)

  • We added the non-pharmacological approach in the manuscript, dealing with omega-3 and exercise therapy.

Reviewer 2 Report

Dear Authors I send you my comments:

1) please add methods

2) line 60-62 please add J Cell Physiol. 2007 Feb;210(2):489-97. doi: 10.1002/jcp.20884.

3) Please add a table and a figure in eas section in order to clarify how stress/depression modulates inflammatory citokines 

Author Response

Reviewer 2

  • please add methods

-> We don’t have relevant methodology. All reference was manually acquired and speculated in terms of authors’ perspectives.

2) line 60-62 please add J Cell Physiol. 2007 Feb;210(2):489-97. doi: 10.1002/jcp.20884.

-> We added it in the manuscript. We appreciate your suggestion of the outstanding reference.

  • Please add a table and a figure in eas section in order to clarify how stress/depression modulates inflammatory citokines 
  • We added a figure in the manuscript. Please refer to the revised manuscript.

Reviewer 3 Report

Dear Authors, I have read your review and I send you my comments:

1) Introduction: please add references and also a figure showing the mechanism

2)  Line 60-62 please cite Int J Immunopathol Pharmacol. 2010 Apr-Jun;23(2):471-9. doi: 10.1177/039463201002300209.

3) Section 2: Please add a Table 

4) PLease add a Table for each section in order to clarify the mechanism of inflammation and the reference

Author Response

Reviewer3

  • Introduction: please add references and also a figure showing the mechanism

-> We added some sentence in the introduction and involved some reference. Figure was inserted in the revised manuscript according to your valuable advice.

  • Line 60-62 please cite Int J Immunopathol Pharmacol. 2010 Apr-Jun;23(2):471-9. doi: 10.1177/039463201002300209.

-> We added it in the manuscript. We appreciate your suggestion of the outstanding reference.

  • Section 2: Please add a Table 

-> For clarification of the whole theory running through the manuscript, we added a figure instead of a table. Please refer to the revised manuscript.

4) PLease add a Table for each section in order to clarify the mechanism of inflammation and the reference

-> We added a figure that clarifies the mechanism according to your valuable advice. Please refer to the revised manuscript.

Round 2

Reviewer 2 Report

Dear Authors,

I have read the manuscript, I have two minor comments:

1) methods: please explain the criteria that you used to select the manuscript, it is a review.  

2) Figure 1: in caption please add what do you mind with MDD

Author Response

We greatly appreciate your kind comments.

We have revised the manuscript according to your comments.

1) methods: please explain the criteria that you used to select the manuscript, it is a review.  

We have added the sentences as follows:

  1. Materials and Methods

The search formulas for ‘depression,’ ‘inflammation,’ and ‘stress’ was used to find the study results for the relationship between stress, inflammation, and depression in last 10 years in the databases of Pubmed, PsychINFO, PsycARTICLES, and Google Scholar. In addition, among the searched ones, the reference lists of review articles were used to search the related articles.

2) Figure 1: in caption please add what do you mind with MDD

We have added the caption as follows:

MDD, major depressive disorder